



# Performance evaluation of an online monitor based on X-ray fluorescence for detecting elemental concentrations in ambient particulate matter

Ivonne Trebs[1], Céline Lett[1], Andreas Krein[2], Erika Matsumoto Kawaguchi[3] and Jürgen Junk[1]

[1]Luxembourg Institute of Science and Technology (LIST), Department of Environmental Research and Innovation, 41, rue du Brill, L-4422 Belvaux, Luxembourg
[2]University of Trier, Faculty VI Spatial and Environmental Sciences, Behringstraße 21, D-54296 Trier, Germany
[3]HORIBA, Ltd., Aerosol Analysis Team, Gas & Fluid Analysis R&D Dept., R&D Division, 1-15-1, Noka, Otsu-shi Shiga 520-0102, Japan

*Correspondence to*: Ivonne Trebs (ivonne.trebs@list.lu)

**Abstract.** Knowledge of the chemical composition of particulate matter (PM) is essential for understanding its source distribution, identifying potential health impacts of toxic elements and to develop efficient air pollution abatement strategies. Traditional methods for analysing PM composition, such as collection on filter substrates and subsequent offline analysis with e.g., inductively coupled plasma mass spectrometry (ICP-MS), are time-

consuming and prone to measurement errors due to multiple preparation steps. Emerging near-real time techniques based on non-destructive Energy Dispersive X-ray Fluorescence (EDXRF) offer advantages for continuous monitoring and source apportionment.

This study characterises the Horiba PX-375 EDXRF monitor by applying a straightforward performance evaluation including (a) limit of detection (LoD), (b) identification and quantification of uncertainty sources, and (c)

investigating and comparing measurement results from three contrasting sites in Luxembourg (urban, semi-urban, rural). We used multi-element reference materials (ME-RMs) from UC Davis for calibration and performed measurements during spring and summer 2023. The LoDs for toxic elements like Ni, Cu, Zn, and Pb were below 3 ng m$^{-3}$ at one-hour time resolution. Higher LoDs were observed for lighter elements (e.g., Al, Si, S, K, Ca). Expanded uncertainties ranged between 5 and 25 % for elemental concentrations above 20 ng m$^{-3}$ and were maximal

for concentrations below 10 ng m$^{-3}$, reaching 60 - 85 %. Elemental analysis revealed S and mineral elements (Fe, Si, Ca, Al) as dominant contributors to PM10. Toxic elements (As, Ni, Pb) were often below the LoD, suggesting minimal exposure risk in the sampled areas. Our results explained on average 51 - 74 % of the gravimetric PM10 mass at the three sites. The study highlights the suitability and importance of the continuous PX-375 particle monitor for future air quality monitoring and source apportionment studies, particularly under changing emission

scenarios and air pollution abatement strategies.



# 1 Introduction

Airborne particulate matter (PM) is composed of organic carbon (OC), elemental carbon (EC), and other chemical species such as sulphate, nitrate, ammonium (SNA), sea salt, mineral compounds, and trace metals. Particle chemical composition is typically determined by measuring the OC, EC, SNA, and sea salt content (Lee and Allen,

2012), but often does not include trace and major elements, which are metals, metalloids, or non-metals. Some trace elements are considered toxic for humans under prevailing high exposure levels (e.g., Pb, Zn, Ni, Cr, As) (Briffa et al., 2020), although EU annual limit values only exist for Pb (0.5 mg m$^{-3}$) (EU Directive 2008/50/EC, currently under revision) and annual target values for Ni (20 ng m$^{-3}$) and As (6 ng m$^{-3}$) (EU Directive 2004/107/EC). Redox active trace metals (e.g., Cu, Zn, Ni, Fe) may be particularly harmful as they can lead to the

generation of reactive oxygen species and subsequent inflammation and oxidative stress (Pant et al., 2015; Daellenbach et al., 2020; Charrier and Anastasio, 2012). Such exposures that mainly originate from tyre and brake wear of heavy exhaust and non-exhaust vehicles can cause serious health problems and ecosystem damages (Baensch-Baltruschat et al., 2020; Beddows and Harrison, 2021; Al Mamun et al., 2020). Under future emission reduction scenarios and more sustainable agricultural practices (e.g., according to the EU Zero pollution action

plan) a decline of the role of SNA in the particle chemical composition is expected. For instance, the European inorganic aerosol pollution load is currently dominated by $NH_4NO_3$ particles (Tang et al., 2021), which will likely decrease with the application of efficient $NH_3$ emission reduction strategies (Guo et al., 2018). Especially natural dust (e.g., composed of the mineral elements Al, Si, Fe and Ca) is expected to play a key role for the non-anthropogenic PM2.5 composition and associated health effects under future air pollution abatement scenarios (Pai

et al., 2022). Future prolonged drought periods and increased aridity in some regions accompanied by extremely dry soils will enhance wind-blown mineral dust emissions (Achakulwisut et al., 2019; Büntgen et al., 2021). Additionally, frequent pyrogenic emissions from wildfires, prescribed burns, and agricultural burning will contribute stronger to the atmospheric aerosol burden (Pai et al., 2022). Resuspension from road dust contributes to the urban PM composition (Maenhaut et al., 2005).

The discontinuous collection on filters and analysis of PM samples using techniques such as inductively coupled plasma mass spectrometry (ICP-MS), inductively coupled plasma optical emission spectrometry (ICP-OES), particle-induced X-ray emission spectrometry (PIXE) and X-ray fluorescence spectrometry (XRF) suffer from several disadvantages, such as high human cost & time, delivering compositional information with a considerable time delay and at low temporal resolution (Tremper et al., 2018) and subsequent difficulties in the analysis of trends

and extreme values. The most common analytical method for discontinuous sampling is ICP-MS, with the primary



drawback of the requirement for sample solubilization, whose effectiveness also depends on the composition of the PM, thereby reducing the representativeness of the sample to be analysed, especially for critical elements to dissolve such as Ti, Cr, and Al (Celo et al., 2010). However, recent works demonstrated the capability of ICP-MS for online analysis of PM (Ji et al., 2022).

Emerging real-time techniques such as laser-induced breakdown spectroscopy determine allow for a detailed physicochemical characteristics of individual particles, providing information on formation mechanisms and fates of atmospheric particles, although, mass concentrations cannot be determined with these techniques (Heikkilä et al., 2024).

Energy Dispersive X-ray Fluorescence (EDXRF) is a state-of-the-art and green-analytical chemistry method to
perform multi-element analysis of airborne PM (Bilo et al., 2024). The method is non-destructive, does not require any sample preparation and use of toxic solvents, and is not affected by the molecular or atomic structure of the elements (Hyslop et al., 2019). The recent technical advancement of EDXRF methods and the increase of detector sensitivities nowadays allows measurements with high temporal resolution for atmospheric/ambient dust monitoring (Furger et al., 2017; Tremper et al., 2018; Asano et al., 2017).

The importance of these near-real time EDXRF methods is currently underrated but will increase in the future as the relevance of spatio-temporal variability assessments and source apportionment studies will likely increase due to the changing chemical particle composition. Currently, two continuous EDXRF online instruments exist, the Xact 625 Ambient Metal Monitor (Sailbri Cooper Inc., US) (Park et al., 2014) and the PX-375 Continuous Particulate Monitor (Horiba, Japan) (Asano et al., 2017). Only few studies have performed online measurements
of particulate element mass concentrations up to now. For instance, the Xact 625 was tested at a rural traffic influenced site in Switzerland for three weeks in summer 2015 (Furger et al., 2017). In addition, it has been evaluated at three contrasting sites (traffic, urban background and industrial) in the UK for several month in 2014, 2015 and 2017 (Tremper et al., 2018). The Horiba PX-375 monitor was deployed in Poland near a moderately inhabited rural area for one month in summer 2018 (Mach et al., 2021). Creamean et al. (2016) used the Horiba
PX-375 for the characterization of particles originating from long-range transport of mineral dust and smoke from forest and grassland fires in the Pacific Northwest to Colorado, US for one week in late summer 2015. Source apportionment of trace elements in PM2.5 was performed using the Horiba PX-375 between April 2014 and April 2015 at five different sites in Beijing (Li et al., 2017). However, in contrast to studies using the Xact 625 (Tremper et al., 2018; Furger et al., 2017) none of the previous studies with the Horiba PX-375 monitor include a rigorous
quality control and uncertainty estimation. The detected metals, metalloids, or non-metals can be used as tracers to identify specific anthropogenic and natural pollution sources and associated health risks (Li et al., 2017; Park et



al., 2014) and a reliable uncertainty estimation provides the basis for source apportionment. A major challenge for these online EDXRF methods is the availability of accurate multi-element reference materials (ME-RMs) for instrument calibration (Bilo et al., 2024). While the suitable Standard Reference Material from the National Institute of Standards and Technology (NIST) was discontinued, the UC Davis Air Quality Research Centre now produces multi-element reference materials with mass loadings corresponding to the range of atmospheric concentrations (Yatkin et al., 2018). Motivated by the need for a detailed performance evaluation of the Horiba PX-375, the four scientific objectives of this paper are: (a) to investigate the limit of detection (LoD) under real environmental conditions, (b) to identify and quantify the uncertainty sources of the PX-375, (c) to evaluate the measurement results collected during spring and summer at three contrasting locations in Luxembourg (rural, semi-urban and urban) and (d) to assess the contribution of detected elements to the total PM10.

## 2 Experimental

### 2.1 Measurement campaigns

With the help of the mobile air quality laboratory (environmental measurement vehicle (EMV)) of LIST (Trebs et al., 2023) an elemental analysis of airborne PM10 was conducted at three different locations in Luxembourg (rural, semi-urban and urban) in 2023 (Table 1, Figure 1). The measurements took place next to the LIST institute building at Belvaux, which is located next to Belval - a district in the southwestern part of Luxembourg (west of Esch-sur-Alzette). The surrounding area includes a mix of residential, commercial, and industrial zones (with steel plants and open slag dump sites in distances of 1 - 5 km). The second location was at the Institut Viti-Vinicole (IVV) in Remich situated in the southeastern part of Luxembourg, within the commune of Remich. The IVV is located within vineyards along the Moselle River. The third sampling location was in the northern part of Luxembourg, situated near the Vianden water reservoir in the Ardennes region with the surroundings dominated by agriculture. Compared to permanent air quality stations the EMV offers an attractive alternative to enhance the spatial and temporal resolution of ambient measurements and identify specific pollution sources. The vehicle is equipped with an X-ray fluorescence analyser (PX-375 Horiba, Japan) (Asano et al., 2017) that was carefully tested, calibrated and quality controlled prior, during and after the field campaigns. Standard meteorological measurements such as air temperature and humidity (3 m), wind speed, and wind direction (6 m) were measured at each location. Simultaneously, PM1.0, PM2.5 and PM10 data were collected with an optical particle counter (GRIMM





Aerosoltechnik EDM 180-F, Ainring, Germany) with a time resolution of one minute at a height of 3 m. These data
120    were averaged and aggregated to match the time resolution of the PX-375.

**Table 1. Overview of sampling locations, periods, and site characteristics for the measurements with the Horiba PX-375 in Luxembourg.**

| Location | Lon \| Lat (WGS84) | Sampling period (2023) | Characteristics |
|---|---|---|---|
| Belvaux | 5.94408 E \| 49.50606 N | 01/03 – 07/03 | Urban, nearby institute, mix of residential, commercial, and industrial areas with occasional construction activities |
| Remich | 6.35483 E \| 49.54535 N | 11/05 – 19/05 | Semi-urban, within wine yard area, occasional constructions around |
| Vianden | 6.14748 E \| 49.94704 N | 29/06 – 06/07 | Rural, near large water reservoir and agricultural sites |

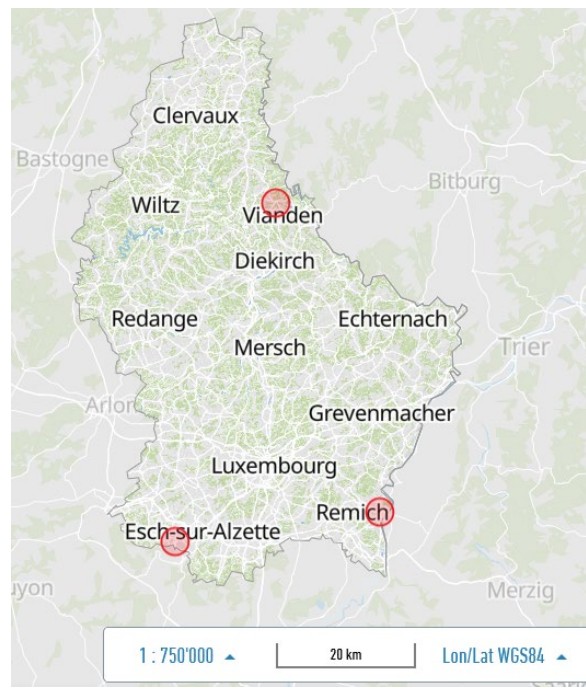

125

**Figure 1: Map of Luxemburg showing the three different sampling locations (Belvaux near Esch-sur-Alzette, Remich and Vianden, map taken from https://map.geoportal.lu).**





## 2.2 Instrument description

The PX-375 was configured to measure ambient concentrations of 15 trace elements in particulate matter (PM10),
namely Fe, Al, Si, Ca, S, Mn, Zn, K, Ti, V, Cr, Ni, Cu, Pb, As. Air was sampled at a flow rate of 16.7 L/min through
US EPA louvered PM10 inlet at a height of 3 m. The inlet is always heated when ambient temperatures are below
40°C. The time resolution can be adjusted from 30 minutes to 24 hours and was set to 120 min (Belvaux, Remich)
and 180 min (Vianden). Although the instrument can measure at lower time resolution, these values were chosen
because relatively low PM10 levels were expected and increased sampling time decrease LoDs. The online PX-
375 monitor applies reel-to-reel filter tape sampling with a none-woven PTFE fabric filter and the spot tape interval
was set to 100 mm. Non-destructive energy-dispersive X-ray fluorescence spectroscopy (EDXRF) analysis is
applied to determine the element concentrations. X-ray pulses are produced with an x-ray tube at two different
photon energy levels (15kV and 50 kV, automatic switching). The analysis time was set to 500 seconds for each
energy level. The total PM10 mass concentration of each sample is determined with beta-ray attenuation. The next
sample is collected on a clean spot of the filter tape while the analyses are performed. The EDXRF unit contains a
metal-oxide-semiconductor (CMOS) camera for sample images allowing the user to control the accurate alignment
of the filter tape such that the particle deposition spot (diameter of 11.5 mm) is exposed to X-ray irradiation
(collimator size is 7 mm). Hence, the manufacturer specifies an acceptable spot diameter tolerance of ±2 mm.

## 2.3 Quality control procedures

To assure measurement quality, X-ray intensity checks, energy calibrations and blank tests were performed after
each filter tape replacement. Additionally, the EDXRF unit was initially calibrated using a blank filter tape and
initially a traceable Standard Reference Material (SRM-2783) from NIST. However, as this SRM was outdated
and the product was discontinued, we re-evaluated the NIST calibration using two freshly produced multi-element
reference materials (ME-RMs) from the UC Davis Air Quality Research Centre (UCD-47-MTL-ME-233 and
UCD-47-MTL-ME-234) with different element loadings (Hyslop et al., 2019; Yatkin et al., 2018). QC of the
calibration was performed by monthly check standard procedures using the blank filter tape and a reference
material, measuring three replicates each time. Additionally, Type A evaluation of uncertainty (Gum, 2008) was
conducted by a check standard procedure including three sequential measurement series (each with n = 10) of both
ME-RMs from UCD and deriving the standard deviation (precision) and bias for each element.

The accurate alignment of the filter tape was checked by manual inspection of the images of the CMOS camera
and ensuring that the sample spot deviation is within ±2 mm. Samples with a deviation of the deposition spot from





the X-ray irradiation position larger than ±2 mm were flagged and treated independently in the uncertainty estimation (see section 3.1.4).

**2.4 Determination of the limit of detection (LoD)**

The limit of detection (LoD) reflects the precision of the instrument response when the concentration of the analyte is zero (Iupac, 1997). This comprises the detection capability (sensitivity and selectivity) of the detector system, and the variability of the system under realistic conditions (including filter tape background). The LoD for each element was calculated based on the conventional definition (Kellner et al., 2004; Iupac, 1997):

$$s_{LoD} = \mu_b + 3\sigma_b \tag{1}$$

with μb the arithmetic mean and σb the standard deviation of the intensity signal of the blank measurements. We measured the same blank spot 10 times, moved filter tape, and measured the new spot 10 times (in total three sets with n = 10 each). The $\sigma_b$ is calculated from the average of the standard deviation of each measurement (n = 10). The concentration at the LoD (cLoD) is then calculated from the analytical sensitivity (slope) of the calibration curve:

$$c_{LoD} = \frac{s_{LoD} - \mu_b}{slope} = \frac{3\sigma_b}{slope} \tag{2}$$

All datapoints below the LoD were set to the respective LoD value for each element.

**2.5 Uncertainty estimation**

For the estimation of the expanded measurement uncertainty, we follow the Guide to the Expression of Uncertainty in Measurement (Gum, 2008). The GUM procedures rely on the assumption that ideally all systematic errors
(biases) have been corrected. The Type A evaluation derives the standard uncertainty by calculating the standard deviation of the mean of a series of independent observations (time-dependent sources of random error) based on check standard measurements (see section 2.3). The Type B evaluation for the PX-375 includes the uncertainty of the ME-RMs provided by the UCD and possible geometry misalignments in the instrument (deviation of the particle deposition spot from the X-ray irradiation position). According to Gum (2008) the combined standard uncertainty
for error sources that are independent from each other is derived by the root-sum-square of the Type A and Type B standard uncertainties. The coverage factor k = 1.96 (95 % confidence level) is used as a multiplier of the combined standard uncertainty to obtain the expanded uncertainty $u_{e_i}$ for each element:

$$u_{e_i} = k\sqrt{\left(\sigma_i{}^2 + u_{rm}{}^2 + u_{spot}{}^2\right)} \tag{3}$$



where $\sigma_i$ is the standard uncertainty to detect the mass of the ith element (Type A uncertainty, expressed as standard

deviation), $u_{rm}$ is the standard uncertainty of the reference material (Type B uncertainty) (Yatkin et al., 2018) and

$u_{spot}$ the spot uncertainty in case of misalignment of the filter tape with the X-ray irradiation beam (Type B

uncertainty). The standard uncertainty of the reference material $u_{rm}$ is derived by dividing the expanded

uncertainty in the RM certificate by the applied coverage factor of two (Yatkin et al., 2018). The $u_{spot}$ was derived

by assessing whether the spot deviation is larger than $\pm 2$ mm in each CMOS picture and by calculating the

erroneous crescent area (see section 3.1.4). This was done by assuming that the particle deposit on the filter tape is

homogenous. The uncertainty of the air flow is $u_Q = 0.2$ L/min (1.2 %) and can be neglected.

## 3 Results and Discussion

### 3.1 Instrument characterisation

**3.1.1** Limit of detection (LoD)

Lowest LoD values ($< 3$ ng m$^{-3}$ at one hour time resolution) were found for the elements Ni, Cu, Zn and Pb (Table

2). The LoDs of the light elements measured with the lower photon energy level of 15 kV (Al, Si, S, K, Ca) were

highest, with maximal values found for Al (100 ng m$^{-3}$) and lowest values for Ca (6.8 ng m$^{-3}$). From the elements

measured with the higher photon energy level of 50 kV (Table 2) Fe showed highest LoDs (8.7 ng m$^{-3}$ at one hour

time resolution). During the blank measurements no signal was detected for As.






**Table 2. Limit of detection (3σ-definition) of air concentrations for the 15 elements detected by the PX-375 using three different time resolutions.**

| element | photon energy level | LoD [ng m$^{-3}$] (1 hour sampling time, 1000 s analysis) | LoD [ng m$^{-3}$] (2 hours sampling time, 1000 s analysis) | LoD [ng m$^{-3}$] (3 hours sampling time, 1000 s analysis) |
|---|---|---|---|---|
| Ti | | 3.0 | 1.5 | 1.0 |
| V | | 7.3 | 3.6 | 2.4 |
| Cr | | 2.9 | 1.4 | 1.0 |
| Mn | | 2.9 | 1.5 | 1.0 |
| Fe | 50 kV | 8.7 | 4.3 | 2.9 |
| Ni | | 1.6 | 0.8 | 0.5 |
| Cu | | 2.4 | 1.2 | 0.7 |
| Zn | | 1.3 | 0.6 | 0.4 |
| As | | - | - | - |
| Pb | | 2.2 | 1.1 | 0.7 |
| Al | | 100.0 | 49.9 | 33.3 |
| Si | | 35.0 | 17.5 | 11.7 |
| S | 15 kV | 13.7 | 6.8 | 4.5 |
| K | | 29.7 | 14.8 | 9.9 |
| Ca | | 6.8 | 3.4 | 2.3 |

The higher LoDs for Si, K and Fe can be explained by the presence of common contaminants in filter raw media
or arising from EDXRF hardware (Hyslop et al., 2022). As some parts inside the PX-375 are made of Al we expect
that the high LoD values were due to marginal contamination during filter tape installation.

In general, the hourly LoDs for Pb, K, Ca, Fe, Zn and Mn are comparable to those already presented for the PX-
375 (Asano et al., 2017) (3σ-definition), while our LoDs for Si and S are two to three times higher than in (Asano
et al., 2017), respectively (Table 2). Moreover, except for Ti and Zn, the hourly LoDs derived in this study are
substantially higher than values presented in Creamean et al. (2016), which were provided by the manufacturer
(Jessie M. Creamean, personal communication) and were thus not determined under real environmental conditions.





In Creamean et al. (2016) the LoD values for Al were also maximal. Compared to the hourly LOD values of elements measured by the Xact in Tremper et al. (2018) (3σ-definition), our LoD value for Si is two times lower while the LoDs of all other elements are higher than those of the Xact. However, it should be noted that the LoD values are not directly comparable as the concentrations of the reference material used for the calibration of the Xact are very high (nearly 1000 times higher than for the PX-375) (Tremper et al., 2018).

**3.1.2** Bias and accuracy

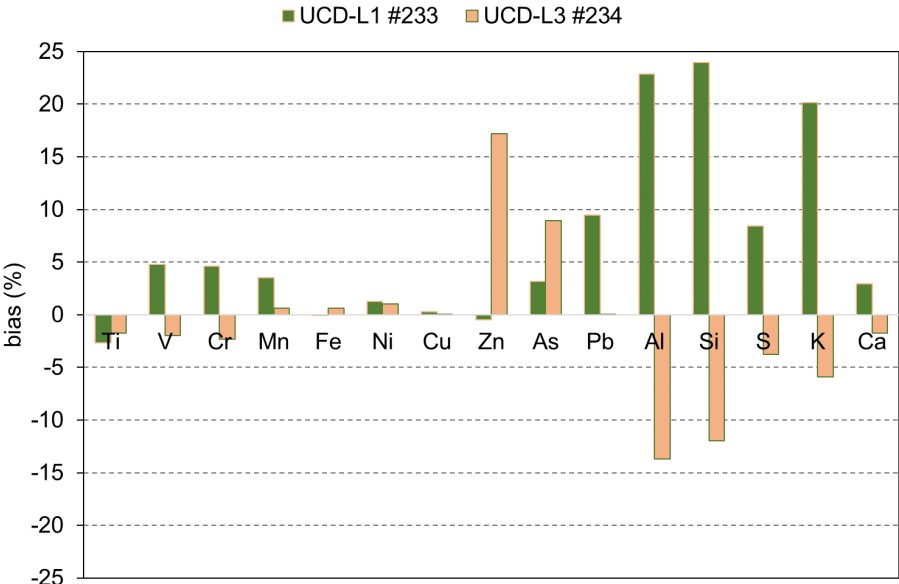

**Figure 2: Instrumental bias (relative average deviation) derived from 30 replicate check standard measurements using two different ME-RMs produced by UCD for 15 elements.**

Systematic uncertainties are defined as non-random variations in the target variable that will decrease the accuracy of a measurement. Figure 2 shows the bias (relative average deviation) of the analysis result for each element for the two different UCD ME-RMs. The bias is below 5 % for the elements Ti, V, Cr, Mn, Fe, Ni, Cu and Ca and is between 5 % and 25 % for the other elements with largest deviations for the lighter elements (e.g., Al, Si, K). For most of the elements, the relative average deviation has a different direction for UCD #233 and UCD #234. This indicates that the observed differences are not systematic, but part of the random uncertainty (Brown et al., 2010). While we consider differences below 5 % to be insignificant and captured by the random uncertainty, the large differences for Al, Si and K are most likely related to the calibration procedure. Figure 3a illustrates the calibration curve for Fe showing a perfect linear fit while the calibration curve for K (Figure 3b) reveals deficiency in linear





fit performance. The residuals indicated by red arrows (+24 % for ME-233 and -5 % for ME-234) are comparable

to the deviations found for K during the check standard procedure (Figure 2), indicating a good reproducibility.

The same was found for Al and Si and we assume that this was related to uncertainty in element loadings of the

RMs. We have included this in the Type-B uncertainty (section 3.1.4). Consequently, no systematic correction of

the results was made.

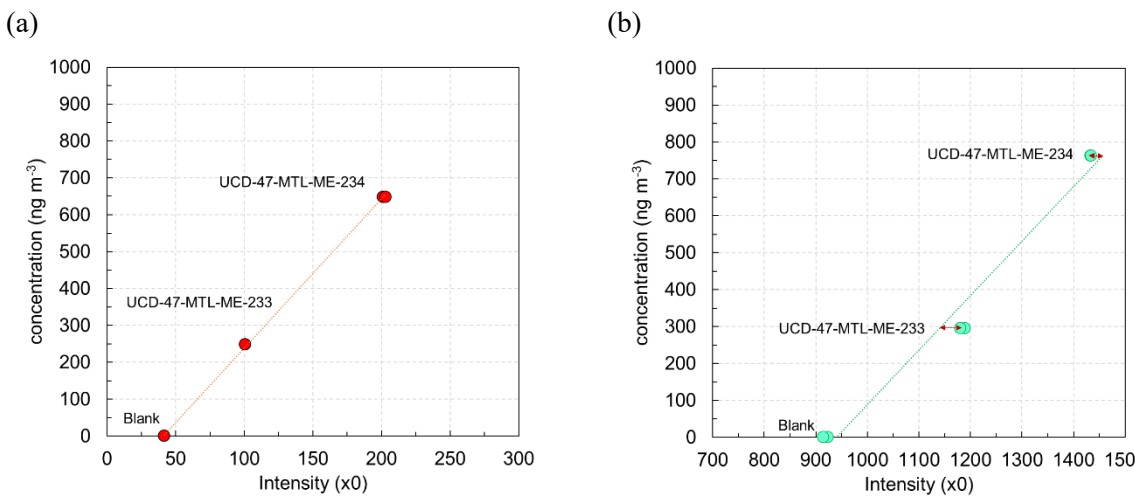

**Figure 3: Example calibration curves (a) for Fe showing a perfect linear relationship ($r^2 = 0.99$) and (b) for**

**K indicating the deviation of the linear curve fit for the data points. Residuals in (b) are indicated by red**

**arrows (+24 % for ME-233 and -5 % for ME-234), standard error of the slope and intercept are 7 % and 10**

**%, respectively ($r^2 = 0.97$).**

In general, uncertainties are expected to be higher for elements with potential for line interferences in multi-element

samples, and from self-absorption effects, for the lightest elements (Si, S, K and Ca) (Furger et al., 2017). The

sensitivity of XRF decreases with lighter elements (Margui et al., 2022). This is also revealed by their higher LoD

values (Table 1).

**3.1.3** Type A uncertainty (precision)

The standard uncertainty (precision as standard deviation divided by mean of repeated measurement results) shown

in Figure 4 is below 4 % for all elements (except Pb). The element loadings of ME-233 are about 2.6 times lower

than for ME-234 and correspond to relatively low atmospheric concentrations typical for a remote region. Figure

4 reveals that analytic precision is not reduced significantly with lower mass loading (UCD-L1 #233) for several





elements. As standard uncertainty $\sigma_i$ for the Type A uncertainty estimation (eq. 3) we have used the average absolute precision for both RMs.

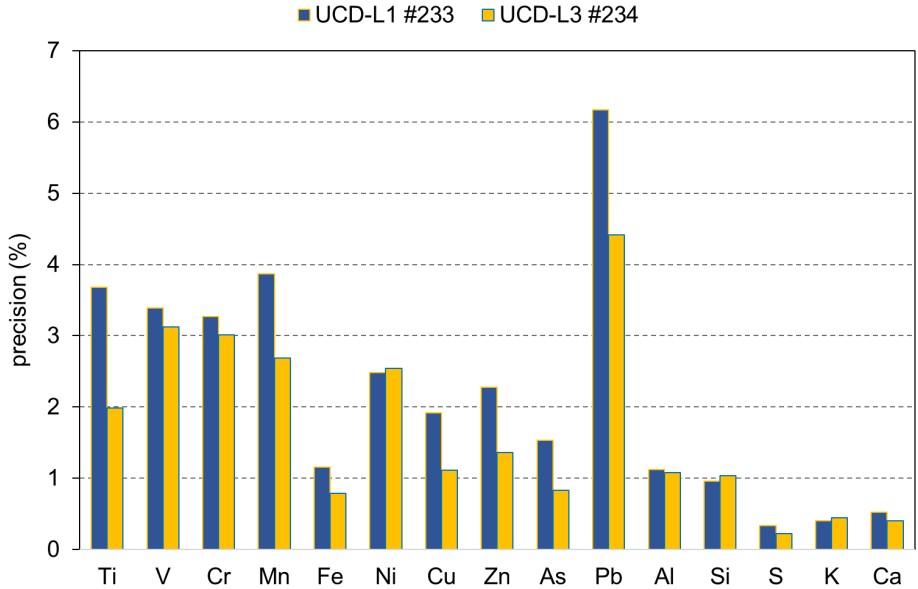

250

**Figure 4: Average relative precision (standard deviation divided by mean) of 30 replicate check standard measurements using two different ME-RMs produced by UCD for 15 elements.**

Relative precision for Fe, Al, Si, S, K and Ca was lowest because their absolute RM element loadings were at least a factor of 10 higher than for the other elements, which is consistent with their expected higher atmospheric
255  concentrations.

**3.1.4** Type B uncertainty

According to the ME-RM certificates all elements have an expanded uncertainty of 10 %, while 20 % was reported only for Ti, As and Si. These expanded uncertainties were converted into standard uncertainties ($u_{rm}$) and included in eq. 3. As we additionally found potential uncertainties of the calibration curve of around 20 % for Al and K that
260  were identified as non-systematic (Figure 3), we increased the Type B RM uncertainty that can also be classified as calibration uncertainty for these two elements to 20 % (see section 3.1.2). The identification of samples for which the deposition spot is outside of specified limit by the manufacturer was done manually (Figure 5).





(a) 03/02/2023 10:00  (b) 15/05/2023 8:00

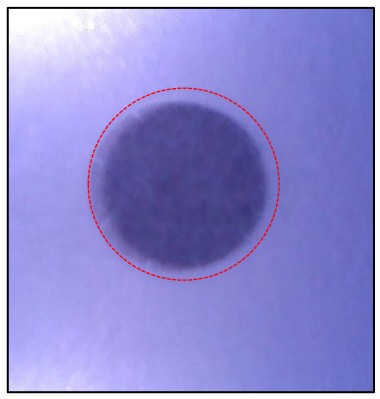 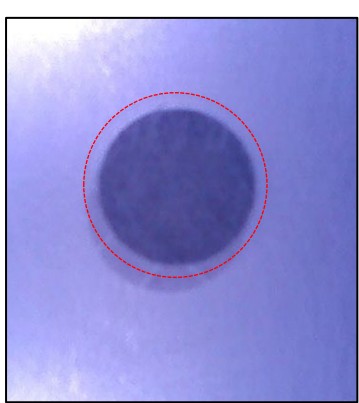

**Figure 5: Sample images of the CMOS camera showing the deviation of the particle deposition spot from the analysis area. The diameter of the spot is 11.5 mm. Collimator size is 7 mm and corresponds to the X-ray exposure spot size. The thin circle around the area indicates a deviation of 1 mm (2 mm are allowed by the manufacturer). Shown are (a) deposition spot well within specified limit and (b) deposition spot near the edge of the specified limit indicating a deviation within the error margin of 2 mm.**

Ideally, this procedure should be done with an automatic image detection and machine learning approach, which is, however, outside the scope of this paper. During all three measurement campaigns, only 1 % of the images were flagged to be outside of the manufacturer limit due to a small misalignment of the filter tape with the X-ray irradiation beam. The erroneous crescent area was calculated for each sample according to standard formulas using the erroneous distance of both circle centres. The determined Type B uncertainty ($u_{spot}$) caused by this minor instrument misalignment is about 2 % and can be neglected in the expanded uncertainty calculations. This result was achieved due to the carful filter tape position adjustment after replacing the filter tape and when we noticed that the deposition spot deviated from the X-ray irradiator. In case this quality assurance is not performed regularly, the Type B uncertainty ($u_{spot}$) can increase to 20 %.

### 3.2 Comparison of NIST and UCD reference materials

The PX-375 was initially calibrated with an outdated NIST SRM (section 2.3). Before re-calibration with the two ME-RMs from UCD, we conducted a check standard procedure with the initial NIST SRM calibration to test the agreement between UCD and NIST standards (Table 3).



**Table 3. Relative differences (average of three values) between measured element loadings of the UCD ME-RMs using the initial calibration with the NIST standard.**

| element | UCD-47-MTL-ME-233 Difference to NIST standard (%) | UCD-47-MTL-ME-234 Difference to NIST standard (%) |
|---|---|---|
| Ti | -140 | -136 |
| V | 89 | 89 |
| Cr | 24 | 23 |
| Mn | 16 | 10 |
| Fe | 6 | -0.3 |
| Ni | -14 | -18 |
| Cu | -7 | -10 |
| Zn | -3 | -10 |
| As | 37 | 30 |
| Pb | -7 | 5 |
| Al | -64 | -9 |
| Si | 35 | 52 |
| S | 57 | 60 |
| K | -13 | 14 |
| Ca | -2 | -9 |

It should be noted that loadings of the NIST standard and UCD standards are different for most elements. NIST loadings were sometimes outside the range of corresponding air concentrations (and quite different to those of UCD), especially for Fe (very high) and S (very low). Although the instrument response is linear, using very low standard materials (as in case of S for the NIST SRM), may introduce uncertainty for higher concentrations. Furthermore, standards that are much higher in concentration than air samples are a limitation (Tremper et al., 2018). We found substantial differences (30-140 %) between NIST and UCD standards for six elements Ti, V, As, Al, Si and S (Table 3). Reasonable agreement was found for all other elements (< 18%) with best agreement for Fe, Zn and Ca (< 10%). Obviously, aging of the NIST standard may have contributed to this discrepancy. However,





the results show the potential drawbacks introduced by using different reference materials and underline the need for universally recognized and certified reference materials (Bilo et al., 2024).

**3.3 Results of the monitoring campaigns with the EMV**

The mean PM10 concentrations during our campaigns are listed in Table 4.

**Table 4. Mean PM10 concentrations for the three different campaigns in Luxembourg during spring and summer 2023.**

| Instrument | Belvaux PM10 [µg m$^{-3}$] | Remich PM10 [µg m$^{-3}$] | Vianden PM10 [µg m$^{-3}$] |
|---|---|---|---|
| Grimm EDM | 25 | 12.5 | 11.1 |
| PX-375 | 16.4 | 11.5 | 10.8 |

For consistency, the contributions to the gravimetric PM10 mass were calculated based on results from the PX-375

only. The 15 measured elements were subdivided into three different groups:

- Mineral airborne elements (MAEs) including Sulphur: Fe, Si, Ca, Al and S
- Hazardous airborne elements (HAEs): Cu, Zn, Cr, As, Ni, Pb
- Other elements (OEs): Mn, K, Ti, V

The toxic elements As, Ni, Pb were often below the LoD and will be excluded in detailed evaluations. Also, the

concentrations of Ti and V were very low.

**3.3.1** Measured time series with expanded uncertainties

The ranges of the relative expanded uncertainties ($u_e$) of the 15 elements are shown in Table 5. The lowest relative $u_e$ values were determined for the elements with the highest ambient concentrations and the lowest Type B uncertainty (e.g., S and Ca). Elements with higher uncertainty of the ME-RM showed lowest $u_e$ values of 18 %

(e.g., Ti, Al, Si, K) (section 3.1.4). For the other elements, the lowest determined $u_e$ values were 5 % (e.g., Cr and Mn) and 9 % (e.g., Fe, Zn). The maximal relative $u_e$ values in each group were due to prevailing extremely low concentrations. Figure 6 shows the dependency of the relative $u_e$ on the measured concentration exemplarily for five elements. The $u_e$ values strongly increase towards lower concentrations (< 10 ng m$^{-3}$), which is particularly obvious for Cr and Zn reaching 50 to 70 % when approaching the LoD. The overall $u_e$ differences for the elements

also apparent at higher concentrations (e.g., for K and Si) are due to their enhanced Type-B uncertainties.





**Table 5. Determined ranges of the relative expanded uncertainties ($u_e$) for the 15 elements.**

| element | $u_e$ (%) |
|---------|-----------|
| Ti | 18 - 69 |
| V | 5 - 70 |
| Cr | 5 - 70 |
| Mn | 5 - 85 |
| Fe | 8 - 40 |
| Ni | 20 - 80 |
| Cu | 10 - 85 |
| Zn | 9 - 42 |
| As | > 100 |
| Pb | > 100 |
| Al | 20 - 23 |
| Si | 19 - 35 |
| S | 9 - 11 |
| K | 19 - 25 |
| Ca | 9 - 16 |





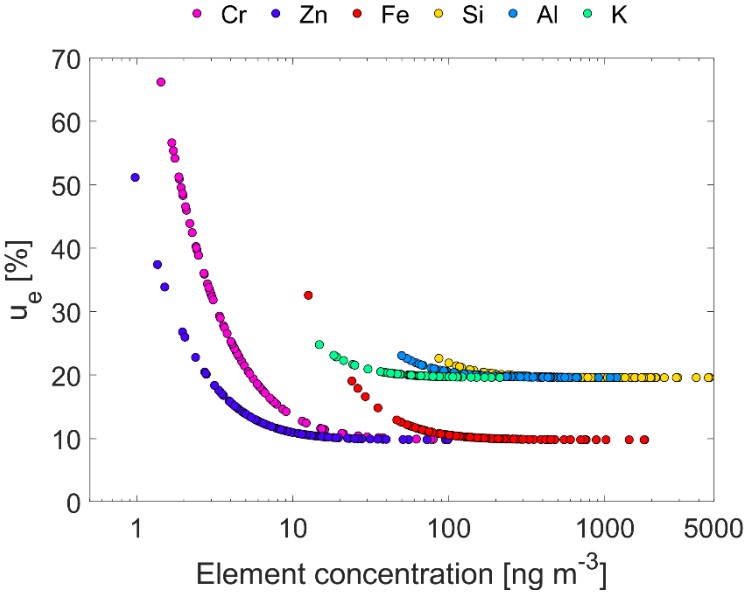

**Figure 6: Relative expanded uncertainty ($u_e$) of the PX-375 exemplarily for Cr, Zn, Fe, Si, Al and K as a function of the element concentration measured during the Remich field campaign in Luxembourg in 2023 (scale of x-axis is logarithmic, time resolution of the instrument was two hours).**

Figure 7 illustrates the results of the three campaigns including expanded measurement uncertainties during spring and summer 2023. S, Zn and K were the most ubiquitous elements identified for MEA, HAE, and OE groups, respectively. Concentrations of the mineral elements Fe, Al, Si and Ca were also elevated at all locations and at least a factor of 10 higher than toxic and other elements. S was the most ubiquitous element in our study and despite the relatively short measurement period the median PM10 concentration of S at the different sites in our study (1190 ng m$^{-3}$) is in agreement with the median PM10 of 1150 ng m$^{-3}$ reported for Europe (Al Mamun et al., 2020). They also report that among major metals/metalloids and MAEs, S has the highest median concentration in PM10, which is in line with our study. The increase/decrease of MAEs with wind speed is a typical pattern (Al Mamun et al., 2020; Tasdemir et al., 2006) and increased MAE values were observed under dry conditions when crustal emissions and resuspension from road dust dominate. For some elements, such as K and S the origin can be anthropogenic (biomass, plant residues) combustion processes, fertilizer application (relevant for vineyards) and crustal erosion, which implies that the diurnal variations can be influenced by mixed source contributions. Concentrations of toxic metals/metalloids (Cu, Zn and Cr) reveal very low concentrations at all locations (Figure 7), and the concentration levels of As, Pb, Ni were mostly below the LoD (not shown). A very recent study of Liu et al. (2024) analysed pan-European results of several trace elements in PM10 and found comparable concentrations





for Cu, Zn and Cr, but substantially higher levels of Ti and Pb compared to our study, although the results are not directly comparable due to the different measurement periods and time resolutions.

The only study in Europe that used the PX-375 was conducted at an urban site in Poland (Mach et al., 2021), where the mean PM10 concentration (20.8 µg m$^{-3}$) was comparable to the urban site Belvaux in our study. However, their results for the elements were contrasting because the concentration of minerals (Fe, Si and Ca) originating from soil borne or resuspended roadside dust were lower, while some of the toxic elements (As, Cu, Ni) were substantially higher than at Belvaux.






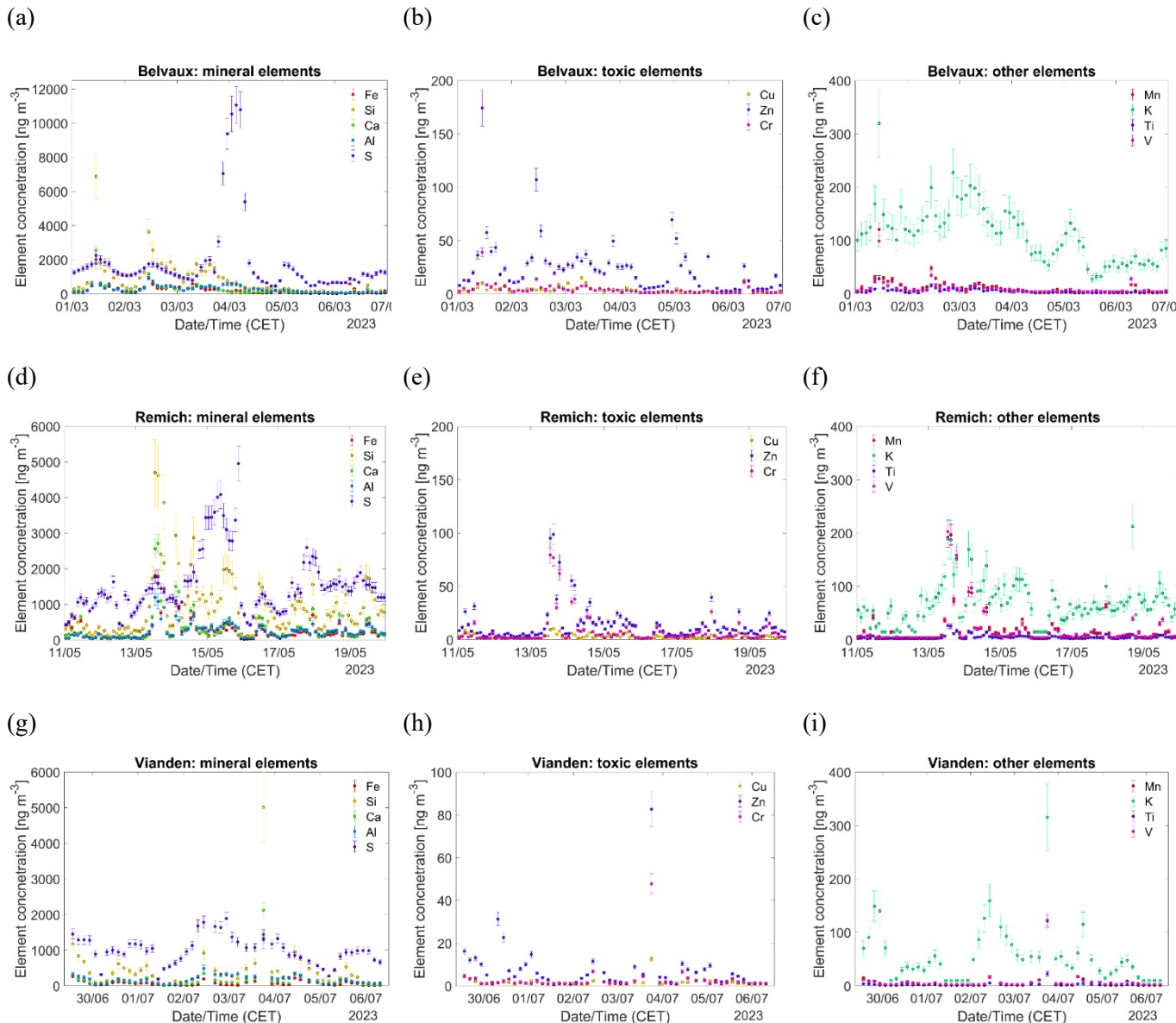

**Figure 7: Measured timeseries of MAEs, HAEs and OEs by the PX-375 during the three field campaigns at (a-c) Belvaux, (d-f) Remich and (g-i) Vianden in Luxembourg during spring and summer 2023. The error bars denote expanded measurement uncertainties $u_{e_i}$ derived for each data point.**



### 3.3.2 Relative contribution of elements

The contribution of the mean detected element concentrations was clearly dominated by S and MAEs (Figure 8).
The largest fraction of the determined mass at all three locations was attributed to S that is mostly associated with particulate sulphate. From all determined elements, S occasionally contributed 80 % (4 March Belvaux and 5 July Vianden), but its contribution also sometimes decreased down to 20 % (13 May Remich). The second largest fraction was contributed by Si, followed by Al, Ca, Fe and K and the sum of minor (mainly toxic) elements (Table 6) (Figure 8). Overall, the elemental contribution to the chemical particle composition was comparable at the three
different sites. No major difference was found between the urban site (Belvaux) and the rural site (Vianden). For the minor (mainly toxic) elements, Table 6 shows that their sum contributed on average less than 2 % to the total of analysed elements, which indicates a very low contribution of non-exhaust traffic emissions and a low exposure to toxic elements. V, Mn, and Zn account for the largest fraction of the analysed minor elements.

**Table 6. Average relative contribution of each minor element to the total of analysed elements by the PX-375 combined for the three sites Belvaux, Remich and Vianden.**

| $Ti_{rel}$ [%] | $V_{rel}$ [%] | $Cr_{rel}$ [%] | $Mn_{rel}$ [%] | $Ni_{rel}$ [%] | $Cu_{rel}$ [%] | $Zn_{rel}$ [%] | $As_{rel}$ [%] | $Pb_{rel}$ [%] |
|---|---|---|---|---|---|---|---|---|
| 0.17 | 0.41 | 0.16 | 0.33 | 0.06 | 0.12 | 0.43 | 0 | 0.11 |

The element contributions represent a complex mix of the geographical origin of emission sources and meteorological conditions and the wind direction measured at a receptor site is not necessarily representative of the
air mass origin and the source distributions (Petit et al., 2017). Additionally, the chemical source profile (i.e. percentage of species with respect to total gravimetric PM mass) varies geographically dependent upon parameters such as traffic (volume and pattern, fleet characteristics), road surface characteristics, and the geology and climate of the region (Pant et al., 2015). Moreover, some elements, such as Fe, Mn, and K can be both of crustal and anthropogenic origin (Fomba et al., 2018).




(a)

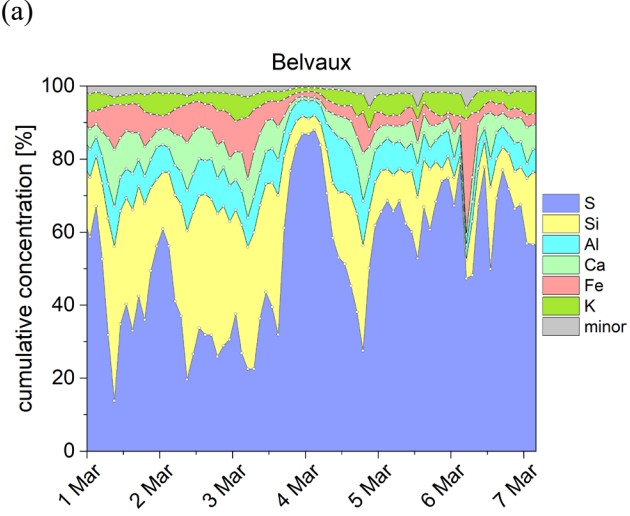

(b)

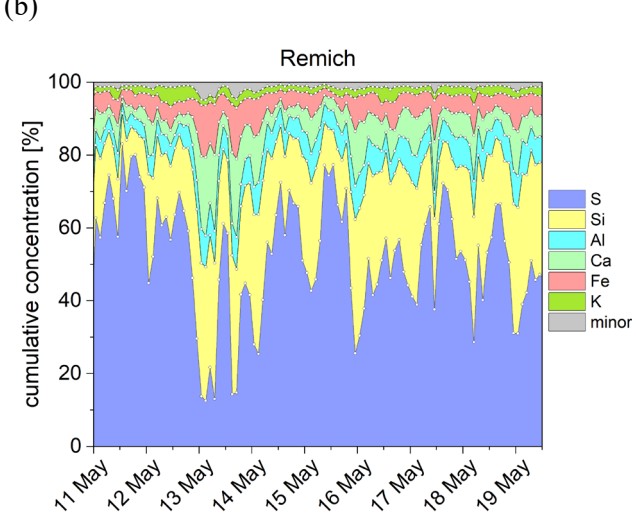



(c)

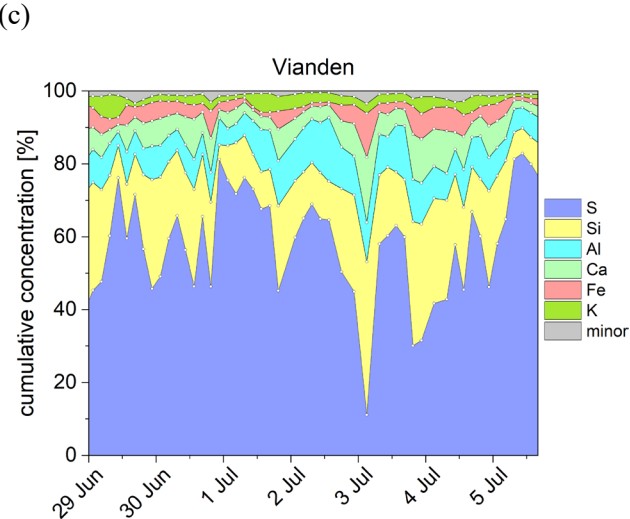

**Figure 8. Relative cumulative elemental concentrations (stacked) indicating the contribution of analysed elements by the PX-375 during the three field campaigns at (a) Belvaux, (b) Remich and (c) Vianden in Luxembourg during spring and summer 2023. "Minor" refers to elements listed in Table 4.**

Figure 9 illustrates that the relative fraction of elemental S increased with the PM2.5/PM10 ratio since most S is attributed to $SO_4^{2-}$ in finer particles related to combustion emissions (Fang et al., 2017). In contrast, the relative fraction of Si decreased with the PM2.5/PM10 ratio due to the fact that mineral elements such as Si are often found in the coarse mode particles due to their high abundance in the crust, but they may also originate from construction activities or resuspended road dust (Maenhaut et al., 2005).





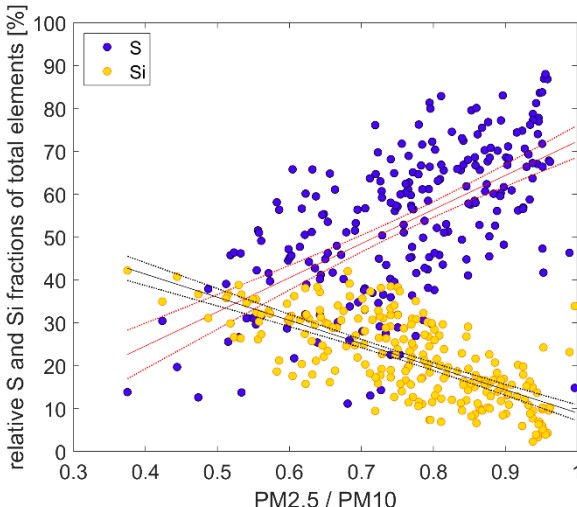

**Figure 9. Relative S and Si fractions of the analysed elements by the PX-375 versus the ratio PM2.5/PM10 (measured with the Grimm EDM 180-F) combined for the three field campaigns at Belvaux, Remich and Vianden in Luxembourg during spring and summer 2023. Included are linear fits with 95 % confidence bounds (red lines for S and black lines for Si).**


The correlations in Figure 9 are stronger for Si ($r^2 = 0.5$) than for S ($r^2 = 0.36$) due to the likely presence of S also in crustal dust, i.e. in coarser particles (Maenhaut et al., 2005). The mobilization of metals (e.g., Cu, Fe) by acidification due to the presence of $SO_4^{2-}$ is likely to remain an important factor in future aerosol oxidation potential and the health effects studies of PM (Fang et al., 2017). It was also recently found that aerosol oxidative potential

was correlated with the elements S, K, Fe, As, Zn, Ca, Mn, and Cu (Molina et al., 2023). The near-real time determination of particulate Si is gaining more attention due to its role as tracer to estimate the secondary aerosol contribution to PM2.5 (Lu et al., 2019) and to quantify primary and secondary particle sources (Yang et al., 2020). Si quantification with conventional ICP-MS methods involves the need for laborious and complex filter extraction or microwave acid digestion procedures (Yang et al., 2002; Bilo et al., 2024). Our results demonstrate that the PX-

375 instrument constitutes a valuable and effective method to detect chemical aerosol properties and element contributions in near-real time, although the measurement uncertainties will impact the source distribution determination of the elements. For instance, positive Matrix Factorization (PMF) solutions are influenced by random uncertainties in the measurement data even when the optimal weighting coefficients and number of factors are set (Chen et al., 2010; Christensen and Schauer, 2008).





### 3.3.3 Contribution to the gravimetric aerosol mass (PM10)

MAEs are largely associated with crustal matter originating from soil, resuspended road-side dust, and construction activities. The total gravimetric mass of crustal matter (CM) in the PM10 samples for the three sites is estimated as (Maenhaut et al., 2005):

$$CM = 1.16 \cdot (1.90\,\text{Al} + 2.15\,\text{Si} + 1.41\,\text{Ca} + 1.67\,\text{Ti} + 2.09\,\text{Fe}) \tag{4}$$

CM accounted for 19 % in Belvaux, 31 % in Remich and 15 % in Vianden to the total PM10 mass, revealing the preponderance of CM at the semi-urban Remich site. On one hand, this was related to some nearby construction activities, but also due to soil emissions during cultivation of the vineyards under relatively dry conditions during the measurement campaign. Although the contribution of crustal matter to the PM10 mass in Remich was exceptionally high, the overall results are comparable to other studies (Fomba et al., 2018; Maenhaut et al., 2005). The fraction of CM in Vianden was relatively high compared to typical rural background sites (Schwarz et al., 2016), revealing that this rural site was exposed to anthropogenic emissions from agriculture and road dust resuspension under dry weather conditions.

We converted the detected average concentrations to gravimetric contributions from brake wear (7.5 [Cu]) and tyre wear (35 [Zn]) (Fomba et al., 2018) resulting in values below 1 %. The detected concentration of S was converted to Sulphate (3 [S]) and its contribution to total PM10 was 31 % at Belvaux, 41 % at Remich and 34 % at Vianden. Hence, our analyses with the PX-375 explained on average 52 %, 74 % and 51 %, of the gravimetric PM10 mass at Belvaux, Remich and Vianden, respectively. The undetermined PM10 mass is associated with elemental carbon, organic matter, ammonium nitrate and sea salt.

### 4 Summary and Conclusions

Although future air pollution abatement scenarios anticipate a decrease of the organic and inorganic aerosol burden due to reduction of fossil fuel emissions, substantial amounts of PM ($> 5$ mg m$^{-3}$) from non-anthropogenic sources are expected to be present (Pai et al., 2022). Moreover, large emissions of non-exhaust PM by e.g., the abrasion of brakes, clutches, and tires of heavy electric vehicles will likely be the dominant traffic sources. Measurements made by the PX-375 cover a large fraction of these semi-natural and non-exhaust particles, mainly including natural dust (crustal elements) and toxic elements attributed with resuspended road dust.

This study presents a comprehensive performance evaluation of the Horiba PX-375 continuous particulate monitor and its application for multi-element analysis of airborne particulate matter (PM10) in Luxembourg. The PX-375 demonstrated low limits of detection (LoD) for several elements, particularly Ni, Cu, Zn, and Pb, with values below



3 ng m$^{-3}$ at one-hour time resolution. Higher LoD values were observed for lighter elements such as Al and Si, likely due to marginal contamination and the inherent characteristics of the EDXRF hardware. The instrument exhibited high precision, with standard uncertainty (Type A) below 4 % for most elements, except Pb. Type B uncertainties, particularly related to the calibration with multi-element reference materials contributed significantly to the overall measurement uncertainty. Elemental analysis during the spring and summer campaigns in Luxembourg revealed that mineral elements (Fe, Al, Si, Ca) dominated the PM10 composition, contributing significantly more than hazardous elements (Cu, Zn, Cr, As, Ni, Pb). Sulphur (S) was the most prevalent element, primarily associated with particulate sulphate, and showed significant variability depending on meteorological conditions and emission sources. The contribution of mineral elements indicated substantial inputs from crustal sources and resuspension of road dust, especially under dry conditions. Toxic elements contributed less than 2 % to the total analysed elements, suggesting low exposure to hazardous components in the study areas.

The PX-375's ability to provide near-real time data on multiple elements enhances our capability for spatio-temporal variability assessments and source apportionment studies. Despite the challenges associated with the calibration and manual control of the accurate filter tape alignment, the PX-375 offers a viable alternative to traditional discontinuous sampling methods, providing valuable insights into the composition and sources of airborne PM with high-time resolution. The study underscores the need for universally recognized and certified reference materials to ensure consistency and accuracy in elemental analysis across different monitoring instruments. The application of the Horiba PX-375 in Luxembourg highlights its potential for enhancing air quality monitoring and source apportionment efforts, contributing to a better understanding and management of atmospheric pollution.

*Data availability.* The data associated with this work will be made available online.

*Author contributions.* Ivonne Trebs: Formal analysis, Visualization, Conceptualization, Validation, Writing - Original draft preparation, Writing - Review & Editing, Project Administration. Céline Lett: Methodology, Validation, Data curation, Formal analysis, Visualization. Erika Matsumoto Kawaguchi: Methodology, Validation, Writing - Review & Editing. Andreas Krein and Jürgen Junk: Funding acquisition, Investigation, Writing - Review & Editing.

*Competing Interest Statement.* All authors declare that they have no competing interests that could have influenced the results reported in this paper.




*Acknowledgements.* This study was funded by the Luxembourg Ministry of the Environment, Climate and Biodiversity and by the Luxembourg Institute of Science and Technology (LIST). ChatGPT was used to improve the closing sentences of the abstract, summary and conclusions.

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
