# Peer review of "Performance evaluation of an online monitor based on X-ray fluorescence for detecting elemental concentrations in ambient particulate matter"

_Atmospheric Measurement Techniques, 2024_

## Referee Comment (RC1)

Review of

"Performance evaluation of an online monitor based on X-ray fluorescence for detecting elemental concentrations in ambient particulate matter"

Summary

The authors describe the use of an online XRF instruments to detect metallic constituents of particulate matter. The instrument is evaluated in controlled, laboratory settings as well as at field sites in Luxembourg with varying anthropogenic influences. Online XRF is a relatively new method, and this work helps fill in gaps of evaluation and near-real time observations. There are a few areas that could be improved prior to publication.

Scientific Questions/Issues

Line 32 – It was surprising for OC and EC to be listed first in the description of PM. First, because they are generally not the dominant components by mass, nor are they chemically defined. The EC and OC parameters are operationally defined depending on the method of analysis. I would recommend moving these after SNA at least.

Line 59 – "difficulties in the analysis of trends" should be elaborated, since several large networks have been deducing atmospheric trends using discontinuous collection on filters for decades.

Lines 83-84 – I recommend including the latest census population estimate with "moderately inhabited rural area", as this can mean very different populations internationally.

Line 130 and subsequent – The ordering of the elements is not conventional. I am guessing that this was done purposely to highlight elements analyzed at the two different energy levels. You may choose to order your elements by atomic number, which aids interpretation of the results and figures.

Line 131 – The US EPA does not manufacture or specify a manufacturer for PM10 inlets. I recommend listing the actual manufacturer for clarity.

Line 167 – Based on the description, it sounds like $\sigma_b$ is calculated from three standard deviations (n=3), one from each set of 10 blank analyses. If that is not correct, maybe more clarity is required here.

Section 2.5 – Are there any uncertainty estimates or potential for stretching of the PTFE membrane? If not, this should be stated.

Table 2 – This table would benefit from an additional column showing LoD estimates from an applicable air monitoring program, such as EMEP or Eurotrac-2.

Line 229 – Many readers will find fault with the use of "perfect" in an experimental result. I suggest replacing with a different adjective (e.g., strong) validated with the metric you used to evaluate it (e.g., $r^2 = 0.99$).

Lines 241-243 – XRF sensitivity to light elements decreases only under insufficient atmospheric control. LoD for light elements can be similar to other elements when using vacuum (< 1mTorr) or even with

helium purge. The choice of detector also matters; Ge and CdTe detectors have inherently low sensitivity to light elements while Si and Si-vortex can have better sensitivity.

Figure 4 – While this view of the precision results is sufficient, it may be better represented with mass loading on the x-axis and viewed as a scatterplot. This may help explain the observed differences.

Line 276 – How was the 20 % Type B uncertainty determined?

Table 4 – The Remich and Vianden instrument comparisons are both close to 10 % while Belvaux is ~ 50 %. This should be discussed or noted for further investigation.

Lines 381-384 and Table 6 – This is not a valid argument or presentation of the data. Trace elements are so called because they contribute little to total mass under normal circumstances. Chronic exposure to elevated levels of toxic metals can induce health effects, in spite of its percent contribution to total PM10 loading. For example, Ni may be three times the EU limit of 20 ng/m$^3$ and only contribute 0.15 % to an acceptable level of PM10 (< 40 µg/m$^3$).

A better comparison would be to calculate relative differences or enrichment with regard to either local geology or the most rural site.

Technical Corrections

Line 65 – "… breakdown spectroscopy determine allow for a …"; maybe just "determine"

Line 137 – The "X" in "X-ray" should be capitalized.

Line 152 – The citation for GUM should be capitalized.

Lines 161 and 164 – The citations for IUPAC should be capitalized.

Figure 2 – There is an excessive amount of space below -15 % and error bars would be useful.

Line 274 – "… achieved due to the carful filter tape …"; maybe "… due to careful filter …"

---

## Author Comment (AC1)

**Response to referee comments**

**Manuscript Number: amt-2024-134**

**Performance evaluation of an online monitor based on X-ray fluorescence for detecting elemental concentrations in ambient particulate matter**
by I. Trebs et al.

**Anonymous Referee #1**

*Reviewer #1: The authors describe the use of an online XRF instruments to detect metallic constituents of particulate matter. The instrument is evaluated in controlled, laboratory settings as well as at field sites in Luxembourg with varying anthropogenic influences. Online XRF is a relatively new method, and this work helps fill in gaps of evaluation and near-real time observations. There are a few areas that could be improved prior to publication.*

**Reply to Reviewer #1:** We would like to thank the referee for the positive evaluation of our manuscript. The proposed revisions will substantially improve the quality of the paper. Please find our detailed responses below:

*Scientific Questions/Issues*

*Line 32 – It was surprising for OC and EC to be listed first in the description of PM. First, because they are generally not the dominant components by mass, nor are they chemically defined. The EC and OC parameters are operationally defined depending on the method of analysis. I would recommend moving these after SNA at least.*

**Reply:** Thanks for the hint. We will change the order of words here.

*Line 59 – "difficulties in the analysis of trends" should be elaborated, since several large networks have been deducing atmospheric trends using discontinuous collection on filters for decades.*

**Reply:** We agree that this statement might be misleading. This will be corrected.

*Lines 83-84 – I recommend including the latest census population estimate with "moderately inhabited rural area", as this can mean very different populations internationally.*

**Reply:** This will be included.

*Line 130 and subsequent – The ordering of the elements is not conventional. I am guessing that this was done purposely to highlight elements analyzed at the two different energy levels. You may choose to order your elements by atomic number, which aids interpretation of the results and figures.*

**Reply:** In fact, the elements listed in Table 2, Table 3 and Table 5 are already sorted according to atomic number, but separately for the two energy levels. We will reverse the order in these tables (15kV before 50kV) so that all elements are sorted by atomic number. However, the timeseries plots cannot be changed as only the elements with comparable ambient concentration ranges can be plotted together.

*Line 131 – The US EPA does not manufacture or specify a manufacturer for PM10 inlets. I recommend listing the actual manufacturer for clarity.*

**Reply:** The PM10 inlet is manufactured by Met One Instruments, Inc. This will be corrected.

*Line 167 – Based on the description, it sounds like $\sigma_b$ is calculated from three standard deviations (n=3), one from each set of 10 blank analyses. If that is not correct, maybe more clarity is required here.*

**Reply:** Yes, this is correct. We will modify the text to make it clearer.

*Section 2.5 – Are there any uncertainty estimates or potential for stretching of the PTFE membrane? If not, this should be stated.*

**Reply:** To our knowledge, there is no uncertainty associated with this. The PTFE could stretch a bit, but its deformation is negligible. We keep the criterion for tear strength (> 8 [N / 25mm].

*Table 2 – This table would benefit from an additional column showing LoD estimates from an applicable air monitoring program, such as EMEP or Eurotrac-2.*

**Reply:** LoDs are sensitive to the integration time of each collection method and cannot be directly compared. Additionally, less than half of the compounds detected here are monitored by EMEP or Eurotrac-2. Also, for some compounds wet/dry deposition is estimated, which is not comparable to the air concentrations in our study.

*Line 229 – Many readers will find fault with the use of "perfect" in an experimental result. I suggest replacing with a different adjective (e.g., strong) validated with the metric you used to evaluate it (e.g., r2 = 0.99).*

**Reply:** We agree and will change the text accordingly.

*Lines 241-243 – XRF sensitivity to light elements decreases only under insufficient atmospheric control. LoD for light elements can be similar to other elements when using vacuum (< 1mTorr) or even with helium purge. The choice of detector also matters; Ge and CdTe detectors have inherently low sensitivity to light elements while Si and Si-vortex can have better sensitivity.*

**Reply:** For modern silicon drift detectors (SDDs) used in XRF, the sensitivity to Al is relatively consistent. For elements such as Na, it depends on the thickness of the window material Be and whether different window materials are used. Indeed, using a vacuum or a helium purge can help in detecting light elements better. These methods reduce the interference from air (or other gases) that can absorb or scatter X-rays. However, the PX-375 does not rely on these methods, which means the detection of light elements like Na might be less effective due to the presence of air. Light elements generally have poor XRF sensitivity compared to transition metals (e.g., Cu). This will be elaborated in more detail in the text.

*Figure 4 – While this view of the precision results is sufficient, it may be better represented with mass loading on the x-axis and viewed as a scatterplot. This may help explain the observed differences.*

**Reply:** This is a good idea and provides a more detailed view. We will replace the plot including this information and revise the discussion of the results in the text.

*Line 276 – How was the 20 % Type B uncertainty determined?*

**Reply:** This is just a hypothetical value in case the deviation of the particle deposition spot from the analysis area becomes very large. We have observed this deviation in previous campaigns before making instrument adjustments. The value is estimated from the erroneous crescent area using the erroneous distance of both circle centres. We will clarify this in the text.

*Table 4 – The Remich and Vianden instrument comparisons are both close to 10 % while Belvaux is ~ 50 %. This should be discussed or noted for further investigation.*

**Reply:** We agree and will make additional remarks in the text.

*Lines 381-384 and Table 6 – This is not a valid argument or presentation of the data. Trace elements are so called because they contribute little to total mass under normal circumstances. Chronic exposure to elevated levels of toxic metals can induce health effects, in spite of its percent contribution to total PM10 loading. For example, Ni may be three times the EU limit of 20 ng/m3 and only contribute 0.15 % to an acceptable level of PM10 (< 40 µg/m3).*

**Reply:** We agree and will make corrections in this part of the text.

*A better comparison would be to calculate relative differences or enrichment with regard to either local geology or the most rural site.*

**Reply:** We have considered the use of enrichment factors while writing the manuscript but noticed that we do not have all data available (e.g., local soil properties). In the revised manuscript, we will include another Table comparing enrichment regarding the most rural site Vianden and some more discussion on this topic.

*Technical Corrections*

*Line 65 – "… breakdown spectroscopy determine allow for a …"; maybe just "determine"*

**Reply:** This is a typo and will be corrected.

*Line 137 – The "X" in "X-ray" should be capitalized.*

**Reply:** This will be corrected.

*Line 152 – The citation for GUM should be capitalized.*

**Reply:** This will be corrected.

*Lines 161 and 164 – The citations for IUPAC should be capitalized.*

**Reply:** This will be corrected.

Figure 2 – There is an excessive amount of space below -15 % and error bars would be useful.

**Reply:** The space below -15 % will be removed in Figure 2. Error bars will be added.

Line 274 – "… achieved due to the carful filter tape …"; maybe "… due to careful filter …"

**Reply:** This is a typo and will be corrected.

---

## Author Comment (AC2)

**Response to referee comments**

**Manuscript Number: amt-2024-134**

**Performance evaluation of an online monitor based on X-ray fluorescence for detecting elemental concentrations in ambient particulate matter**
by I. Trebs et al.

**Anonymous Referee #2**

Reviewer.–8¿The manuscript describes the use and evaluation of an online energy-dispersive X-ray fluorescence (EDXRF) detector, the Horiba PX-375, for elemental analysis of ambient particulate matter. The team characterized the performance of the detector, including its limit of detection and measurement uncertainties, and compared the field measurements.

This online EDXRF technique offers advantages for non-destructive, near-real-time, and continues measurements, as well as source apportionment. A comprehensive study and understanding of the detector's performance is highly desired. I would recommend accepting the manuscript with minor edits.

**Reply to Reviewer #2:** We would like to thank the referee for the positive evaluation of our manuscript. The proposed revisions will substantially improve the quality of the paper. Please find our detailed responses below:

*Line 160-171: Equations (1) and (2) need clarification. In particular, is the standard deviation term the average of three standard deviations? Also, how is the calibration curve derived?*

**Reply:** The details for these equations are described in the text, line 167: "The $\sigma_b$ is calculated from the average of the standard deviation of each measurement (n = 10)". The reviewer is correct, the standard deviation is the average of three $\sigma_{bi}$. We will add some more details in the text.

The calibration curve is derived by plotting the blanks and concentrations of the UCD standards (ME-233, ME-234) with the instrument response (intensity x0) for each standard. This results in scatter plots with three data points and linear regression analysis is used to obtain the equation to derive the mass of the samples (normalized by the sampled air volume). An example is shown in Figure 3. The convention is to plot the instrument response data on the y-axis and the values for the standards on the x-axis. This is due to the assumption that the errors in the instrument response values (due to random variation) are greater than those in the values assigned to the standards. However, we reversed this plotting scheme as this is not the case for our data because the standard uncertainty of the ME-233, ME-234 (10-20%) is larger than the precision (random variation) of the instrument. We will add some more details in the text.

*Line 175-177: Is the standard uncertainty calculated from the mean of a series of observations, or from the standard deviation of the observations, or from several means of several series of observations?*

**Reply:** In lines 152-154 it is stated that the **"**Type A evaluation of uncertainty was conducted by a check standard procedure including three sequential measurement series (each with n = 10) of both ME-RMs from UCD and deriving the standard deviation (precision) and bias for each element." We will add some more information in the text directly related to this calculation.

*Line 198-199: Are 50 kV and 15 kV the energies of the incident photons radiating the samples? Or are they the incident beams hitting some targets, generating photons that then excite the samples. My understanding from the manuscript (Line 137) is that they are the incident photon energies directed at the sample. However, in that case, I am not sure why 50 kV is used instead of 15 kV for Fe. Fe is excited with 15 kV and has a higher cross section at 15 keV than 50 keV. In fact, many elements listed in the 50 keV section of Table 2 should be excited with 15 keV incident photons. It may be helpful to include more instrument details.*

**Reply:** Yes, they are the incident photon energies directed at the sample. We clarify this in the manuscript. When the tube voltage is 15 kV, the maximum energy of the X-rays irradiated to the sample is 15 keV, and the intensity of the X-rays (7.11 keV or higher) that can excite Fe is weak, so Fe cannot be excited efficiently. A tube voltage of 50 kV results in better excitation efficiency.

*Line 240: Does "self-absorption" here refer only to the signal absorption by the particles themselves, or does it include more general signal absorption, such as by the air path, detector window, etc.? Corrections for the air path and window thickness should be implemented in the data quantification. The impart of particle absorption can be estimated by the size of the particles.*

**Reply:** It refers only to signal absorption by the particles. This, however, cannot be quantified at the exact size of the individual particles is unknown. Elements are quantified considering the conditions of the detector and its surroundings (air path and window thickness). We will add some more details about this in the revised manuscript.

*Lastly, I suggest adding a description of the spectrum analysis, including, for example, spectrum fitting and peak identification.*

**Reply:** When the energy is assigned to the horizontal axis and the pulse counts to the vertical axis, the values obtained by the multi-channel analyzer are shown as a spectrum on the LCD. The constituent elements are identified by the peak positions on the spectrum and quantitatively analyzed by the peak height. We will add some more details about this in the revised manuscript.

*Technical corrections:*

*Line 82, should be "several months"*

**Reply:** This typo will be corrected.

---

## Author Response (AR2)

**Editor comment:**

I just have one minor comment to consider. On lines 472-473, would the authors expect much sea salt in Luxemburg? Perhaps rephrase this to state that these are possible contributions to the undetermined PM10, depending on the origins and transport pathways of the air masses arriving at each of the sites. Primary biological particles should also be included in this list. The authors should consider citing this work with a recent review on atmospheric aerosol composition that delves into PM10 source contributions.

**Response:**

Dear Editor,

Thank you very much for this important remark. We have improved the text accordingly and added two references.

Best regards,

Ivonne Trebs